# The Hatching Time of Broiler Chickens Modifies Not Only the Production Traits but Also the Early Bacteriota Development of the Ceca

**DOI:** 10.3390/ani13172712

**Published:** 2023-08-25

**Authors:** Nikoletta Such, Kornél Schermann, László Pál, László Menyhárt, Valéria Farkas, Gábor Csitári, Brigitta Kiss, Kesete Goitom Tewelde, Károly Dublecz

**Affiliations:** 1Institute of Physiology and Nutrition, Hungarian University of Agriculture and Life Sciences, Georgikon Campus, Deák Ferenc Street 16, 8360 Keszthely, Hungary; such.nikoletta.amanda@uni-mate.hu (N.S.); schermannkornel2000@gmail.com (K.S.); pal.laszlo@uni-mate.hu (L.P.); farkas.valeria@uni-mate.hu (V.F.); csitari.gabor@uni-mate.hu (G.C.); kiss.brigitta.gyongyi@phd.uni-mate.hu (B.K.); tewelde.kesete.goitom@phd.uni-mate.hu (K.G.T.); 2Institute of Technology, Hungarian University of Agriculture and Life Sciences, Georgikon Campus, Deák Ferenc Street 16, 8360 Keszthely, Hungary; menyhart.laszlo@uni-mate.hu

**Keywords:** broiler chickens, parent flock age, hatching time, production traits, bacteriota composition, ceca

## Abstract

**Simple Summary:**

Chicken meat is one of the main protein sources of animal origin worldwide and its production has been increasing steadily. These chickens have huge growth potential and a short production period, so all the factors that affect the vitality of day-old birds are getting more and more important. An other important issue in animal production is the decreased use of antibiotics to stabilize the bacteriota composition of the gut. In this article, the effects of parent flocks with different ages and the hatching time of the chickens were evaluated according to the production parameters, hatchability, and gut bacteriota composition of chickens. From the results, it can be concluded that the live weight of day-old chickens is crucial because it determines the growth rate of birds for the whole fattening period. Different parent flocks had no effect, but the hatching time modified the bacterium composition of the ceca at day 11. The reason for this could be the differences in the bacteriota colonization in the hatcher and the feed and water access between the early- and late-hatched chickens.

**Abstract:**

This trial was carried out to find out the effects of the parent flock and hatching time of broiler chickens on the production traits and bacteriota development of animals. Two sets of 730 hatching eggs were collected from two different parent flocks with ages of 25 and 50 weeks. In the hatchery, both groups were divided into two subgroups: those hatched during the first 10 and the subsequent 10 h of the hatching window. A feeding trial was carried out afterwards, using the four treatments in six replicate floor pens and feeding commercial starter, grower, and finisher diets that contained all the nutrients according to the breeder’s recommendations. The day-old chickens of the older parent flock and those hatched later were heavier, and this advantage remained until the end of the production period. The different ages and origins of the parent flocks failed to modify the microbiological parameters of the chicken’s ceca; however, the hatching time significantly influenced the different bacteriota diversity indices: the late-hatched chickens showed higher *Bacteroidetes* and lower *Firmicutes* and *Actinobacteria* abundances at day 11. These treatments resulted in differences in the main families, *Ruminococcaceae*, *Lactobacillaceae*, and *Bacteroidaceae.* These differences could not be found at day 39.

## 1. Introduction

It is well known that the stability of the intestinal microflora is crucial in the efficient use of nutrients in farm animals. The initial contact with the hen contributes to the development of the intestinal flora [1]. This relationship with the hens is missing in the intensive production systems in the hatcheries. Therefore, the development of intestinal flora, for example, in broiler chickens, is more accidental and influenced by several environmental factors [1,2,3]. Another important issue that can modify the vitality and production potential of the chickens is the length of the hatching window. This means that the chicks in the hatchery do not hatch at the same time. While some animals spend only a short time in the brooder, other chicks can spend even 20–24 h without access to feed and water. Our hypothesis was that besides the differences in the physiological and metabolic status of the early- and late-hatched chickens, the time spent in the brooder also means more time for the early-hatched animals to be colonized with the spore-forming bacteria on the eggshell surface. Little is known about this effect on the early development of the intestinal microbiota [4,5].

It is well known that the weight of the eggs and the weight of day-old chickens increase with the age of the parent stock [6]. Chicks of the older hens are heavier and exhibit significantly higher growth rates than birds from younger flocks [7,8]. During the development of the chicken embryo, it has been estimated that more than 90% of the total energy requirement is derived from yolk lipids [9]. According to the results of Hamidu et al. [10], the breed and age of the parent stock influence the daily embryonic metabolism, which is almost exclusively fuelled by lipids. It is also well known that the older broiler breeder hens produce eggs with higher eggshell pore numbers than the young ones. Changes in the eggshell structure also modify the conductance of oxygen and carbon dioxide across the eggshell [8,11,12]. However, it is not known whether this eggshell characteristic could have an influence on the development of the intestinal microbiota in the embryo or day-old chicken.

The aim of this study was to investigate the effects of the age of the parent stock and the hatching time on the performance parameters and cecal microbiota composition of broiler chickens.

## 2. Materials and Methods

### 2.1. Egg Collection and Hatching

Eggs were collected from a 25- and a 50-week-old Ross 308 flock from different farms. A total of 730 eggs were collected per farm and transported in air-conditioned trucks (16–18 °C) to the hatchery (Gallus Ltd., Devecser, Hungary). There was a one-day difference between the eggs’ arrival to the hatchery; that is, the eggs of the younger layers arrived one day earlier. The average egg weight was 53.4 and 69.1 g of the young and old flocks, respectively. The eggs were disinfected via formalin fumigation with paraformaldehyde at a concentration of 7 g/m^3^. Before incubation, the eggs were stored at 16–18 °C and 75–80% humidity. The hatching was started on the seventh day after laying in Petersime Bio Streamer 24S-type pre-hatching machines. The temperature, relative humidity, CO_2_ content of the air, and the rotation of the eggs were automatically performed by the hatching machine according to the standard hatching protocol, starting with 37.9 °C and 94% humidity, which was reduced at day 18 to 35.7 °C and 74%. On day 18, infertile, dead, damaged, and rotten eggs were selected with candling. All the fertile eggs were then vaccinated against infectious bursal disease (IBD) (Cevac Transmune) with automatic equipment (Embrex Inovoject, Zoetis Inc., New York, NY, USA). After inoculation, the machine automatically moved the eggs to the brooding trays. The environmental parameters of the brooders were also controlled automatically according to the normal hatching protocol. The temperature at the start was 36.7 °C, which was reduced to 35.3 °C on the last day of brooding. The humidity in the brooder was 82% in the beginning; this value increased to 89% on the second day and declined again to 82%. The collection of the early-hatched animals happened in the period between 481 and 491 h, and that of the late-hatched chickens between 492 and 502 h of incubation time. The average hatching time was 489.4 and 493.9 h in the early- and late-hatched groups, respectively. After the selection, the day-old chickens were vaccinated against infectious bronchitis (Cevac Bron 120 L, Ceva-Phylaxia, Budapest, Hungary) and Newcastle disease (Cevac Vitapest) and transported immediately. They arrived in the experimental farm within 2 h.

### 2.2. Animal Experiment and Treatments

A floor pen trial was conducted at the experimental farm of the Institute of Physiology and Nutrition, Hungarian University of Agriculture and Life Sciences (Georgikon Campus, Keszthely, Hungary). The animal experiment was approved by the Institutional Ethics Committee (Animal Welfare Committee, Georgikon Campus, Hungarian University of Agriculture and Life Sciences) under the license number MÁB-5/2022. All husbandry and euthanasia procedures were carried out in accordance with the Hungarian Government Decree 40/2013 and in full consideration of animal welfare ethics.

From the total 280 chickens of each parent flock and hatching time group, 144 birds were selected with similar live weight and transported to the experimental farm. A total of 576 chickens were allocated into 4 treatment groups with 6 replicate floor pens of 24 chickens. The net surface of pens was 1.5 m^2^, which meant 16 chickens per m^2^ stocking density. The following treatments and abbreviations were used: young parent stock and early-hatched chickens (YE); young parent stock and late-hatched chickens (YL); old parent stock and early-hatched chickens (OE); and old parent stock and late-hatched chickens (OL).

During the experiment, computer-controlled housing and climate conditions were maintained according to the breeder’s recommendations [13]. The housing temperature reduction steps were as the follows: d 1–2: 30 °C; d 3–5: 29 °C; d 6–8: 28 °C; d 9–11: 27 °C; d 12–14: 26 °C; d 15–17: 25 °C; d 18–20: 24 °C; d 21–30: 23 °C; d 24–26: 22 °C; and d 27–39: 21 °C. Feed and water were available ad libitum throughout the whole experiment. Corn–soybean-based diets were fed in all groups. The nutrient content of the diets met the requirements of Ross 308 broiler chickens [13]. Three phases were used during fattening. The starter diets (0–10 days) were fed in mash; the grower (11–24 days) and finisher feeds (25–39 days) were fed in pelleted form. The composition and nutrient contents of the diets are shown in Table 1.

### 2.3. Measurements and Sample Collection

During the 39-day-long fattening period, the bodyweight (BW) of all the animals was measured at day 0 and at the end of each feeding phase. Feed intake (FI), bodyweight gain (BWG), and feed conversion ratio (FCR) were calculated on a pen basis for each phase and for the entire period. On days 11 and 39, 2 chickens per pen were selected randomly, slaughtered, and digesta samples were collected. Cecum chymus samples were collected from the left sac. The luminal contents were homogenized with sterile cell spreaders and about 2 g sample was taken into a sterile container. All samples were immediately snap-frozen and stored at −80 °C until analysis. Before DNA extraction, the samples of two birds of the same pen were pooled.

### 2.4. DNA Extraction, 16S rRNA Gene Amplification and Illumina MiSeq Sequencing

The extraction of the bacterial DNA was carried out using the AquaGenomic Kit (MoBiTec GmbH, Göttingen, Germany) and further purified using KAPA PureBeads (Roche, Basel, Switzerland) according to the manufacturer’s protocols [14]. The genomic DNA was investigated using a Qubit 3.0 Fluorometer with a Qubit dsDNA HS Assay Kit (Thermo Fisher Scientific Inc., Waltham, MA, USA). Bacterial DNA was amplified with tagged primers covering the V3–V4 region of the bacterial 16S rRNA gene [15]. Polymerase chain reactions (PCR) and DNA purifications were performed according to Illumina’s demonstrated protocol (Illumina Inc., San Diego, CA, USA, 2013). PCR product libraries were defined and qualified using High Sensitivity D1000 ScreenTape on TapeStation 2200 instrument (Agilent Technologies, Santa Clara, CA, USA). Equimolar concentrations of libraries were pooled and sequenced on an Illumina MiSeq platform using a MiSeq Reagent Kit v3 (600 cycle; Illumina Inc.) 300-bp read length paired-end protocol. Raw sequence data of 16S rRNA metagenomics analysis were deposited in the National Center for Biotechnology Information (NCBI) Sequence Read Archive under the BioProject identifier PRJNA996958.

### 2.5. Bioinformatics and Statistical Analyses

Bacteria were identified via the analysis of the V3–V4 region of the 16S rRNA gene using Illumina MiSeq platform. Sequences were analyzed using Quantitative Insights into Microbial Ecology 2 (QIIME2), version 2020.2 software package [16]. Sequences were filtered based on quality scores and the presence of ambiguous base calls using the quality-filter q-score options. Representative sequences were found using a 16S reference as a positive filter, as implemented via the deblur denoise-16S method. Sequences were clustered into Operational Taxonomic Units (OTUs) using VSEARCH algorithm open-reference clustering based on a 97% similarity to the SILVA reference database [17]. Alpha diversity metrics (Chao1, Shannon, and Simpson) and beta diversity metrics (Bray–Curtis dissimilarity) were estimated using Qiime2-diversity and Microbiomanalyst online software [18] after samples were rarefied to 1000 sequences per sample. To examine the differences in the microbial community structure between samples, SPSS statistical software version 23.0 (IBM Corp. Released 2015) was used. To verify the significance of bacterial community, an analysis of similarities (ANOSIM) and calculations were performed with 999 permutations.

The results of hatchability were evaluated using the Fisher exact test of the R Statistic programme [19]. The production traits and microbiota composition data were evaluated using the two-way analysis of variance of the SPSS software (version 23.0—IBM Corp. Released., 2015), using the hatching time and the age of the parent flocks as the main factors. The microbial composition at different taxonomical levels were compared using a two-way ANOVA test with Benjamini–Hochberg false discovery rate correction (FDR *p*-value). Normality of data (Shapiro–Wilk test) and homogeneity of variance (Levene’s test) were checked prior to statistical testing. Statistical significance was defined as *p* < 0.05, whereas a *p*-value between 0.05 and 0.10 was considered as a trend.

## 3. Results

### 3.1. Hatchability

The age of the parent flock did not influence the percentage of fertile eggs, damaged eggs, or rotten eggs; however, the egg hatchability of the younger parent flock was significantly higher (Table 2).

### 3.2. Production Traits of Birds

Both the hatching time and the age of the parent flock resulted in significant differences in the hatching weight of the chickens. The hatching weight of the chickens of the older parent flock and those hatched later was significantly higher (Table 3). The advantage of the day-old chickens from the older parent stock remained until the end of the fattening period and resulted in a significantly higher growth rate and better feed conversion ratio. The differences in the production traits between the early- and late-hatched chickens were less, but the cumulative weight gain of the late-hatched chickens was also significantly higher.

### 3.3. Microbiota Analyses

In this study, from all 48 samples, a total of 799.166 good-quality 16S rRNA reads were available for analysis after quality filtering. The average sequence number was 16.649 (min: 3136; max: 24,402). These sequences were assigned to 701 OTUs at 97% similarity using the open approach.

#### 3.3.1. Alpha and Beta Diversity

The Shannon and Simpson diversity indices demonstrated that the microbiota of early-hatched birds was more diverse than that of the late-hatched chickens at day 11 (Table 4). No such differences in the alpha diversity were found at day 39 (Table 5).

Beta-diversity based on principal coordinate analysis (PCoA) ordination using the Bray–Curtis dissimilarity matrix showed a significantly different (PERMANOVA R-squared = 0.087 *p* = 0.004) bacterial community structure between the chickens of the two hatching times (Figure 1A) at 11 days of age. This difference was not visible at the age of 39 days (R-squared = 0.056 *p* = 0.18; Figure 1B). No significant differences due to parent flock age at day 11 (R-squared = 0.051 *p* = 0.239; Figure 1C) or day 39 (R-squared = 0.034 *p* = 0.643; Figure 1D) were revealed.

#### 3.3.2. Taxonomic Composition of Cecal Microbiota at Phylum Level

At day 11, six, and at day 39, eight phyla were identified in the cecal contents of the birds (Table 6 and Table 7). At both time points, *Firmicutes* was the major dominant phylum in the cecum followed the *Bacteroidetes* and *Tenericutes*. Minor phyla were *Proteobacteria*, *Cyanobacteria*, and *Actinobacteria*. At 39 days, *Epsilonbacteraeota* and *Euryarchaeota* appeared in a small proportion. The age of the parent flocks failed to influence the bacteriota composition of the ceca at phylum level. On the other hand, at day 11, the time of hatching affected the abundances of three phyla significantly (Table 6). *Firmicutes* was present at a significantly higher abundance in the early-hatched birds (88.488–77.39%; *p* = 0.048), while as a trend, the abundance of phyla *Bacteroidetes* (7.982–20.108%; *p* = 0.056) and *Actinobacteria* (0.086–0.053%; *p* = 0.089) were higher in the late-hatched animals. These differences had disappeared by day 39 (Table 7).

#### 3.3.3. Taxonomic Composition of Cecal Contents at Family Level

The detailed treatment effects are shown in Appendix A. Both main factors failed to result in significant differences in the ratio of the bacterial families. However, similarly to the changes in phylum level, as a tendency, hatching time resulted in differences in the ratio of the main families. The families above 1% abundance, as affected by the hatching time, are shown on the taxa bar plot (Figure 2). There were four major families (*Ruminococcaceae*, *Lachnospiraceae*, *Bacteroidaceae*, and *Lactobacillaceae*) and ten minor families above 1%. As the taxa bar plot shows, the family abundance patterns of the early- and late-hatched chickens are different. At this age, in the ceca of the early-hatched chickens, the abundance of *Ruminococcaceae* and *Lactobacillaceae* was 7% and 4.6% higher, respectively, while that of *Bacteroidaceae* was 12.2% lower than in the 39-day-old animals. The *Lachnospiraceae* family was not influenced by the hatching time. In accordance with the diversity and phylum results, the differences between the families had disappeared by day 39.

## 4. Discussion

According to the literature data, there are differences in the hatchability of the eggs between the young and old parent flocks [20,21,22,23]. In the study of Roque et al. [22], hatchability and viability (hatchability of fertile eggs) were lower in the younger, 27–31-week-old flock due to the increased early- and late embryonic mortalities. This is consistent with what can be found in the breeders’ management manual [14]. In our case, the fertility and the embryonic death rate was not worse in the younger flock, and the hatchability was even higher. Similarly to our results, Abudabos et al. [23] found higher mid-term dead embryos from older hens. Egg storage before hatching could also be a factor, which depresses egg albumen Haugh units (HU) and chick quality [21]. This effect is greater in old, 45-week-old breeding hens.

It is also well-known that the breeder’s age influences the weight of the egg and the day-old chicken [7,8,24], and this bodyweight effect can persist until slaughter [25].

After the chick hatches and the remaining yolk complex is withdrawn into the abdominal cavity, the lipid assimilation and metabolism of the yolk continues and is sufficient to adequately maintain the chick for several days after hatching [9]. The time of hatching resulted in a significantly higher hatching weight for birds hatched later. The reason behind this may be the weight loss of “early” hatched chicks and the greater depletion of their glycogen stores [26]. In addition, the late-hatched chickens are more mature in development at the time of hatching [5]. There are plenty of results available on the effects of feed and water deprivation on the metabolism and viability of young broiler chickens [27,28,29]. The novelty of this result is that no artificial deprivations were used, but only the effects of the hatching window during a normal hatchery practice were measured. Similarly to the parent flock age effect, the hatching time also affected the final bodyweight of the animals, which was, significantly, 122 g higher in this trial. Of course, immediate feed and water allowance of the early-hatched chickens could compensate for the weight loss. It highlights the importance of the novel hatchery feeding technologies that reduce the stress and improve the adaptation ability of the day-old chickens [30]. Besides the higher growth potential of the late-hatched chickens, the higher variance in the day-old weight results also increase the live weight variance of the flocks later on, which impairs feed conversion.

The aim of this study was also to find out if the different parent flocks or the hatching time of the chickens can modify the early development of the cecal microbiota. The potential impact of the parent flocks could be either the different eggshell structure [31,32] or the differences in the environmental and farm conditions, and in this way, the vertical microbiota transfer from the hens to the egg [33,34,35,36].

No significant differences have been found in the diversity indices and microbiota composition between the chickens of the two parent flocks at any time interval. Since the diets of the two flocks were identical, it means that the bacteriota transfer via the eggs is determined mainly by genetic factors. The effects of the other farm conditions are low, probably since the bacteria cannot get through the eggshell [31,32], and most of the bacteria are killed during the disinfection in the hatcheries [37].

On the other hand, the hatching time caused several significant changes in the cecal bacteriota in this trial. The time interval between hatch and first feeding affects the development and function of intestinal tract [38]. The development of the intestinal tract consists of the increase of the total length and weight of the intestine, as well as the length and area of the intestinal villi [4]. The immune and thermoregulatory system of poultry undergo significant physiological changes too. The lipids of egg yolk are the primarily source of energy during the early post-hatch period [39,40]. Several factors influence the residual yolk weight at hatch, especially egg size and incubation temperature, while the breeder hens’ age affects the nutrient composition of the residual yolk [40]. The transport from the yolk sac into the intestine was observed up to 72 h after hatching [41]. In addition, it was also described that the yolk utilization was more rapid in fed than in fasting birds, suggesting that the transport of yolk through the intestine could be increased by the greater intestinal activity found in fed chicks. It is known that in the first days after hatching, the deprivation of feed slows down the gut development, as is reflected by lower gut weight, shorter length, lower enzyme activity, altered villi and crypt cell density, and lower crypt depth and height in the short and long term [5].

In mammals, it has been proven that microbes exist in different regions of the placenta and that microbial DNA can be transferred horizontally from mother to fetus through the placenta [42,43]. In addition, however, the structure and succession of the gut microbiota is influenced by many factors, such as the method of delivery, the birth environment, and dietary habits [44,45]. In the case of birds, the eggshell forms a barrier to microbial transfer to the embryo, but also provides an important protection against environmental pathogens [46]. In hatcheries, the newly hatched chicks have only limited contact with the hen’s microbiota [2]. This is mainly restricted to the transfer of microbes to their offspring during the egg formation process [47]. This is important because the host’s microbes can prevent the infections, increase hatchability, and can be beneficial in the early bacteriota development [48,49].

Several studies reported that the delay in access to feed may affect the microbiota development. A huge increase in microorganisms occurs in the chicken’s intestine after the first ingestion of feed [50,51,52]. According to our results, the early-hatched birds, which were longer without feed, had more diverse cecal microbiota than the late-hatched birds. The reason for this difference is not known. One explanation could be that the early-hatched chickens had more contact with eggs in the brooder baskets and could be colonized with some spore-forming bacteria that survive the disinfection in the hatchery. Disinfection reduces the bacterial load on the eggshell surface from more than 10^4^ CFU to about 10^3^ CFU [37]. On the eggs, the dominating phyla are *Firmicutes*, *Actinobacteria*, *Fusobacteria*, and *Tenericutes*. After the fumigation process in the hatchery, the ratio of *Firmicutes* decreases, that of *Actinobacteria* increase, and, as new phyla, *Proteobacteria*, *Bacteroidetes*, and *Cyanobacteria* are present [37]. Disinfection results in more diverse egg surface bacteriota composition at lower taxonomic levels. We could not prove this hypothesis, since no changes in the spore-forming bacterial groups have been found due to the differences in the hatching time. Usually, early feed access increases the bacterial diversity in the intestine [52], but we could not find results specifically on hatching time-induced changes.

In our study, a higher abundance of *Bacteroidetes* and a lower abundance of *Firmicutes* and *Actinobacteria* was found in the late-hatched chickens. *Actinobacteria* is one of the four main phyla of the cecal microbiota, and although its abundance is low, the bacteria of this phylum play an important role in maintaining intestinal homeostasis since they can use a wide variety of complex polysaccharides [53,54]. Several studies proved the importance of the *Bacteroidetes* and *Firmicutes* ratio in the different gut segments [55,56,57]. The frequency of the *Bacteroidetes* phylum is very variable (10–57%) in birds of slaughter age. An important difference between the two phyla is that while members of *Firmicutes* express fucose isomerase, members of *Bacteroidetes* express xylose isomerase [55]. Members of *Bacteroidetes* are present mostly in the distal intestine, where they participate in supplying the host with energy obtained from feed through the fermentation of otherwise indigestible polysaccharides [58]. They are also important participants of the cross-feeding mechanisms, providing substrates for lactic acid-producing bacteria, and can provide extra energy if fibrous diets are fed [53,59,60]. In addition, the secretion of antimicrobial peptides is also a characteristic feature of members of the phylum, which also supports the positive ecological function of *Bacteroidetes* [56,61]. Furthermore, representatives of *Bacteroidetes* also produce propionate, resulting in a beneficial balance between maintaining homeostasis and producing sufficient energy from available nutrients [55]. In the case of the late-hatched birds, the higher frequency of *Bacteroidetes* phylum may be because the early-hatched animals depend longer only on the nutrients of the yolk sac. It contains mostly fats and protein and only low amounts of carbohydrates [27]. However, the late-hatched birds got access to digestible and indigestible carbohydrates-containing feed sooner, which could promote the colonization of *Bacteroidetes* in the ceca. Li et al. [62] found the opposite, wherein 24 h- and 48 h-long feed deprivation resulted in significantly higher *Bacteroidetes* and *Actinobacteria* abundances. However, the results are not fully comparable because of the differences in the treatments.

In the first few days, the dominance of the *Firmicutes* phylum is more beneficial since many of its members are butyrate producers. The production of butyrate in the young chicken’s ceca is important because of the high demand for butyrate for the intensive growth of intestinal cells and to exclude members of the first-colonizer potential pathogens; for example, *Clostridia* and *Enterobacteriacea* [58,63,64].

Although at the family level, the differences between the early- and late-hatched chicken’s microbiota were not significant, the abundance of the families *Lactobacillaceae* and *Ruminococcaceae* belonging to the *Firmicutes* phylum decreased, while that of family *Bacteroidaceae* increased in the late-hatched chickens. Members of the *Lactobacillaceae* family produce lactic acid, which is the main substrate for several members of the *Ruminococcaceae* family, which use lactate as a substrate to produce butyrate and caproic acid [65]. Because of this cross-feeding mechanism, the close correlation between *Lactobacillaceae* and *Ruminococcaceae* is therefore not surprising. The other main butyrate-producing family, *Lachnospiraceae*, was not affected by the treatments. In contrast, the members of the *Bacteroidaceae* family contain several genes encoding cellulose and complex polysaccharides-degrading enzymes [61,66,67] and produce, besides propionate and butyrate, different oligosaccharides [56,68].

## 5. Conclusions

From the results of this experiment, it can be concluded that besides the well-known factors, the time of hatching also has a significant effect on the live weight of day-old chickens. The late-hatched chickens are heavier, and they can maintain this advantage even to the end of the fattening. Therefore, shortening the hatching window or using early feeding techniques decreases the live weight variance of day-old birds and later of the flock. Lower variance in the live weight means a more accurate nutrient supply and a better feed conversion ratio.

The other novel finding of this trial was that hatching time also modifies the early bacteriota development in the ceca. At day 11, the alpha diversity of the early-hatched chickens was higher than that of the late-hatched birds, suggesting more bacterial contact of the early-hatched birds in the brooder. Since there was no big difference in the access to feed between the early- and late-hatched chickens, this result means that the bacteria on the eggshell that survive the disinfection process could also have an impact on the colonization of day-old chickens.

The ceca of late-hatched chickens contained more bacteria belonging to the phylum *Bacteroidetes* and less of *Firmicutes* at day 11. The exact mechanism behind this shift in bacteriota is not known, but the higher abundance of *Bacteroidetes* is positive, since these bacteria play an important role in fiber degradation and supporting substrates in the context of the so called cross-feed mechanism to the lactic acid and butyric acid-producing microbes.

## Figures and Tables

**Figure 1 animals-13-02712-f001:**
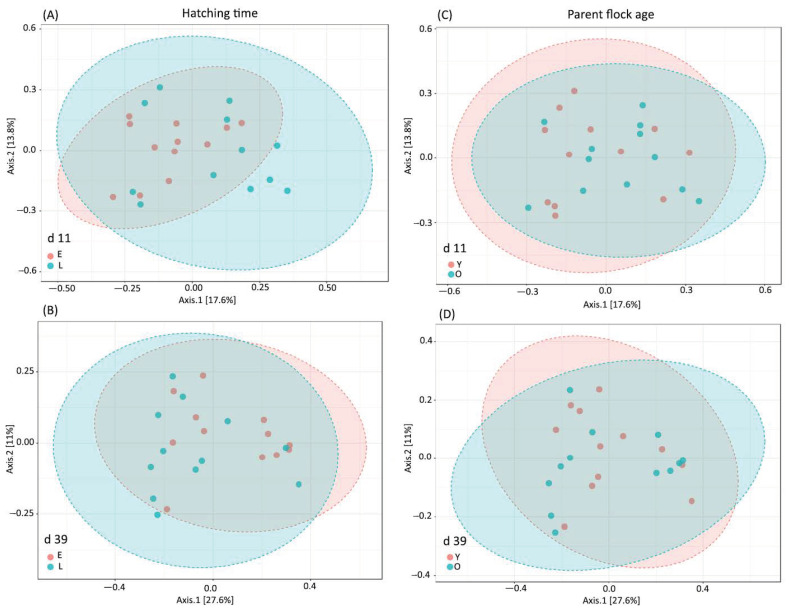
Principal coordinate analysis (PCoA) based on Bray–Curtis dissimilarity matrix on sampling sites: (**A**) hatching time effect at day 11 (E—early-hatched; L—late-hatched); (**B**) hatching time effect at day 39; (**C**) parent age effect at day 11 (Y—young parent; O—old parent); (**D**) parent age effect at day 39. The percentage of variation explained by each PCoA is indicated on the axes with Bray–Curtis dissimilarity. To verify the significance of the bacterial community, permutational analysis of variance (PERMANOVA) calculations were performed. The differences were considered significant at a level of *p* ≤ 0.05.

**Figure 2 animals-13-02712-f002:**
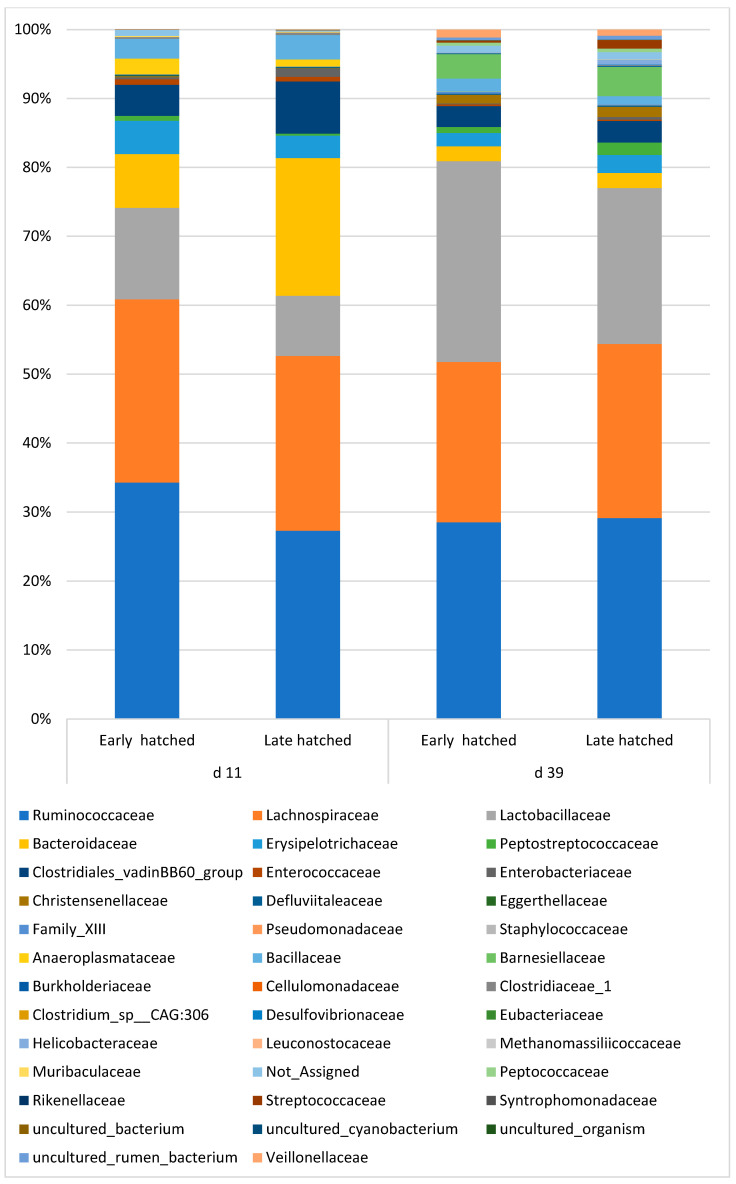
Effects of hatching time on the cecal microbiota composition at the family level.

**Table 1 animals-13-02712-t001:** The composition and analyzed nutrient content and of experimental diets (g/kg as fed).

Composition of the Diets	Starter	Grower	Finisher
Corn	391.3	424.8	479.4
Wheat	100.0	100.0	100.0
Extracted soybean meal	407.0	374.0	321.0
Sunflower oil	51.0	60.0	64.0
Limestone	16.5	13.9	12.0
MCP	13.2	11.0	8.9
l-Lysine	4.1	2.7	2.1
dl-Methionine	4.0	3.2	2.9
l-Threonine	1.4	0.8	0.6
l-Isoleucine	0.3	0.1	0.1
l-Arginine	0.3	0	0
l-Valine	1.0	0.6	0.6
NaCl	3.0	3.0	3.0
NaHCO_3_	1.0	1.0	1.0
Premix ^1^	5.0	4.0	4.0
Xylanase ^2^	0.2	0.2	0.2
Phytase ^3^	0.1	0.1	0.1
Coccidiostat ^4^	0.6	0.5	0
Sum	1000	1000	1000
Measured nutrient contents			
AMEn (MJ/kg)	12.48	12.42	13.16
Crude protein	22.3	19.2	18.6
Crude fat	7.0	8.0	8.4
Crude fibre	4.2	4.1	4.1
Ca	1.04	0.94	0.88
P (total)	0.63	0.61	0.53

^1^ Premix was supplied by UBM Ltd. (Pilisvörösvár, Hungary). The active ingredients in the premix were as follows (per kg of diet): retinyl acetate—5.0 mg; cholecalciferol—130 μg; dl-alpha-tocopherol-acetate—91 mg; menadione—2.2 mg; thiamin—4.5 mg; riboflavin—10.5 mg; pyridoxin HCL—7.5 mg; cyanocobalamin—80 μg; niacin—41.5 mg; pantothenic acid—15 mg; folic acid—1.3 mg; biotin—150 μg; betaine—670 mg; Ronozyme^®^ NP—150 mg; monensin-Na—110 mg (only grower); narasin—50 mg (only starter); nicarbazin—50 mg (only starter); antioxidant—25 mg; Zn (as ZnSO_4_·H_2_O)—125 mg; Cu (as CuSO_4_·5H_2_O)—20 mg; Fe (as FeSO_4_·H_2_O)—75 mg; Mn (as MnO)—125 mg; I (as KI)—1.35 mg; Se (as Na_2_SeO_3_)—270 μg. ^2^ NSP digesting enzymes, beta 1-4, endo-xylanase enzyme—Econase XT, AB Vista, Marlborough, Wiltshire, SN8 4AN. ^3^ Quantum Blue (Panadditív Kft. 2040, Budaörs, Hungary); ^4^ Maxiban, Elanco Clinton Laboratories, Clinton, IN, USA.

**Table 2 animals-13-02712-t002:** Effects of the parent flock’s age on the different hatchability characteristics.

	Fertile Eggs (%)	Damaged Eggs (%)	Rotten Eggs (%)	Hatched from Fertile Eggs (%)	Hatched of All Eggs (%)
Old parent flocks	95.6	0.56	5.83	93.8 ^b^	89.2 ^b^
Young parent flocks	97.5	0.14	4.44	95.4 ^a^	92.9 ^a^
SEM	1.58	0.29	2.17	2.33	3.30
*p*-values	0.168	0.374	0.236	0.047	0.016

The hatchability parameters were evaluated with Fisher exact test. The differences were considered significant at a level of *p* ≤ 0.05. ^a,b^ means with different superscripts of the same column are significantly different. Data were expressed as means ± SEM.

**Table 3 animals-13-02712-t003:** Effects of treatments on the bodyweight, feed intake, the feed conversion ratio, and weight gain of broiler chickens.

Parent Flock Age		Young		Old		Parent Flock Age		Hatching Time		SEM	*p*-Values		
Hatching Time		Early	Late	Early	Late	Young	Old	Early	Late		Parent Flock Age	Hatching Time	Interaction
Bodyweight (g)	d 0	35.2	36.9	47.7	49.7	36.4 ^b^	48.3 ^a^	41.7 ^b^	42.9 ^a^	1.254	0.000	0.000	0.826
	d 10	230.2	228.1	310.2	302.9	229.1 ^b^	306.5 ^a^	270.2	267.8	8.546	0.000	0.439	0.668
	d 24	1112.9	1150.3	1362.5	1397.2	1131.6 ^b^	1379.8 ^a^	1237.7	1273.7	27.449	0.000	0.057	0.943
	d 39	2513.9	2581.5	2808.5	2984.9	2547.7 ^b^	2896.7 ^a^	2661.2 ^b^	2783.2 ^a^	42.587	0.000	0.004	0.156
Feed intake (g/day)	starter	286.2	289.8	303.6	316.0	288.0 ^b^	309.8 ^a^	294.9	302.9	3.488	0.001	0.145	0.413
	grower	1137.7	1153.2	1320.5	1317.5	1145.4 ^b^	1319.0 ^a^	1229.1	1235.3	24.079	0.000	0.857	0.788
	finisher	2162.3	2147.9	2332.8	2362.9	2155.1 ^b^	2297.9 ^a^	2197.5	2255.4	30.024	0.012	0.279	0.179
	sum	3586.3	3590.9	3857	3996.5	3588.6 ^b^	3926.8 ^a^	3721.6	3793.7	51.528	0.000	0.364	0.395
Bodyweight gain (g)	starter	194.9	194.5	262.7	262.0	194.7 ^b^	262.4 ^a^	228.8	228.3	7.417	0.000	0.375	0.738
	grower	895.4	918.7	1052	1099.3	907.0 ^b^	1075.6 ^a^	973.7 ^b^	1009.0 ^a^	19.335	0.213	0.209	0.420
	finisher	1387.6	1447.5	1456.5	1574.5	1417.6 ^b^	1515.5 ^a^	1422.1 ^b^	1511.0 ^a^	17.933	0.873	0.084	0.456
	cum.	2478.5	2560.8	2771.3	2936.0	2519.4 ^b^	2853.6 ^a^	2624.7 ^b^	2748.4 ^a^	41.021	0.028	0.082	0.678
FCR (g/g)	starter	1.47	1.49	1.16	1.20	1.48 ^a^	1.18 ^b^	1.31	1.35	0.036	0.000	0.919	0.980
	grower	1.32	1.24	1.29	1.24	1.26	1.22	1.26	1.22	0.015	0.000	0.030	0.468
	finisher	1.60	1.50	1.57	1.54	1.52	1.51	1.54	1.49	0.015	0.001	0.001	0.226
	cum.	1.49	1.40	1.41	1.39	1.42 ^a^	1.37 ^b^	1.41	1.38	0.012	0.000	0.003	0.289

cum.—cumulative. Data were expressed as means ± SEM. The production parameters were evaluated with two-way ANOVA, using the hatching time (early; late) and the age of parents (young; old) as the main factors. The differences were considered significant at a level of *p* ≤ 0.05. ^a,b^ means with different superscripts of the same column are significantly different.

**Table 4 animals-13-02712-t004:** Alpha diversity indices of the cecum chymus at day 11.

		Parent Flock Age				*p*-Values	
	Hatching Time	Young Parent	Old Parent	Average (Hatching Time)	Hatching Time	Parent Flock Age	Interaction
Chao 1	Early	175.952	193.232	184.592	0.481	0.814	0.406
	Late	178.026	168.328	173.177			
	Average (Age)	176.989	180.780				
Shannon	Early	3.720	3.862	3.791 ^a^	0.018	0.708	0.187
	Late	3.546	3.295	3.420 ^b^			
	Average (Age)	3.633	3.578				
Simpson	Early	0.951	0.955	0.953 ^a^	0.011	0.483	0.354
	Late	0.918	0.889	0.904 ^b^			
	Average (Age)	0.935	0.922				

Alpha diversity indices were compared using two-way ANOVA, using the hatching time (early; late) and the age of parents (young; old) as the main factors. The differences were considered significant at a level of *p* ≤ 0.05. ^a,b^ means with different superscripts of the same column are significantly different.

**Table 5 animals-13-02712-t005:** Alpha diversity indices of the cecum chymus at day 39.

		Parent Flock Age				*p*-Values	
	Hatching Time	Young Parent	Old Parent	Average (Hatching Time)	Hatching Time	Parent Flock Age	Interaction
Chao 1	Early	380.614	367.537	374.076	0.100	0.951	0.438
	Late	335.548	350.850	344.474			
	Average (Age)	360.130	358.552				
Shannon	Early	4.318	4.189	4.254	0.800	0.889	0.429
	Late	4.174	4.264	4.382			
	Average (Age)	4.252	4.229				
Simpson	Early	0.958	0.951	0.954	0.741	0.667	0.295
	Late	0.941	0.960	0.966			
	Average (Age)	0.951	0.955				

Alpha diversity indices were compared using two-way ANOVA, using the hatching time (early; late) and the age of parents (young; old) as the main factors. The differences were considered significant at a level of *p* ≤ 0.05.

**Table 6 animals-13-02712-t006:** Effects of hatching time and the age of parent flocks on the cecal microbiota composition at the phylum level at 11 days of age.

Phylum (%)		Parent Flock Age			FDR *p*-Values		
	Hatching Time	Young Patent	Old Parent	Average (Hatching Time)	Hatching Time	Parent Flock Age	Interaction
*Firmicutes*	Early	84.830	92.146	88.488 ^a^	0.048		0.290
	Late	83.304	71.476	77.390 ^b^			
	Average (Age)	84.067	81.811		0.625		
*Bacteroidetes*	Early	9.917	6.047	7.982 ^B^	0.056		0.170
	Late	12.458	27.759	20.108 ^A^			
	Average (Age)	11.188	16.903		0.290		
*Proteobacteria*	Early	0.161	0.429	0.295	0.104		0.122
	Late	1.932	0.400	1.166			
	Average (Age)	1.046	0.415		0.269		
*Actinobacteria*	Early	0.077	0.096	0.086 ^A^	0.089		0.913
	Late	0.042	0.063	0.053 ^B^			
	Average (Age)	0.059	0.080		0.250		
*Cyanobacteria*	Early	1.513	0.085	0.799	0.187		0.313
	Late	0.171	0.000	0.086			
	Average (Age)	0.842	0.043		0.342		
*Tenericutes*	Early	3.502	1.198	2.350	0.157		0.896
	Late	2.093	0.302	1.197			
	Average (Age)	2.798	0.750		0.100		

The microbiota composition at phylum level were compared using two-way ANOVA, using the hatching time (early; late) and the age of parents (young; old) as the main factors. The differences were considered significant at a level of *p* ≤ 0.05. ^a,b^ means with different superscripts of the same column are significantly different. ^A,B^ means with different superscripts of the same column show a trend.

**Table 7 animals-13-02712-t007:** Effects of hatching time and the age of parent flocks on the cecal microbiota composition at the phylum level at 39 days of age.

Phylum (%)		Parent Flock Age			FDR *p*-Values		
	Hatching Time	Young Patent	Old Parent	Average (Hatching Time)	Hatching Time	Parent Flock Age	Interaction
*Firmicutes*	Early	91.905	93.253	92.579	0.596		1.483
	Late	91.120	90.007	91.630			
	Average (Age)	91.513	91.630		1.064		
*Bacteroidetes*	Early	6.626	5.036	5.831	0.471		1.186
	Late	6.684	6.648	6.666			
	Average (Age)	6.655	5.842		1.132		
*Proteobacteria*	Early	0.257	0.201	0.229	0.303		0.872
	Late	0.553	0.591	0.572			
	Average (Age)	0.405	0.396		0.952		
*Actinobacteria*	Early	0.028	0.032	0.030	0.533		1.385
	Late	0.024	0.072	0.048			
	Average (Age)	0.026	0.052		0.844		
*Cyanobacteria*	Early	0.638	1.160	0.899	0.545		0.955
	Late	0.927	1.490	1.209			
	Average (Age)	0.783	1.325		0.622		
*Tenericutes*	Early	0.279	0.290	0.284	0.485		0.937
	Late	0.232	0.181	0.206			
	Average (Age)	0.255	0.235		1.167		
*Epsilonbacteraeota*	Early	0.248	0.000	0.124	0.468		0.943
	Late	0.437	0.987	0.712			
	Average (Age)	0.343	0.493		1.046		
*Euryarchaeota*	Early	0.014	0.024	0.019	0.988		0.855
	Late	0.019	0.020	0.019			
	Average (Age)	0.017	0.022		1.185		

The microbiota composition at phylum level were compared using two-way ANOVA, using the hatching time (early; late) and the age of parents (young; old) as the main factors. The differences were considered significant at a level of *p* ≤ 0.05.

## Data Availability

All data generated or analyzed during this study are included in the published article (and its Appendix A). Raw sequence data of 16S rRNA meta-genomics analysis are deposited in the National Center for Biotechnology Information (NCBI) Se-quence Read Archive under the BioProject identifier PRJNA996958.

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
