# Peer review of "The Hatching Time of Broiler Chickens Modifies Not Only the Production Traits but Also the Early Bacteriota Development of the Ceca"

_animals, 2023, doi:10.3390/ani13172712_

Round 1
Reviewer 1 Report
This is an interesting paper, but lack of details in the experimental design did not give a chance to fully access the quality of the paper.
1. The weight of eggs were not presented. Why? This the main parameter for incubation. The eggs from the old parent stock could be substantially heavier.
2. The details of hatching timing were not presented. What exactly was the difference in time of access to feed between the groups?
3. At which conditions chick after hatching were stored before placement for growth. How long that period was?
4. Why data on microbial population of egg shell after fumigation were not studied and presented? It is necessary to present literature data on this subject.
5. How long eggs were stored before incubation. Since they were collected at different farms. The storage time probably was different between group.
6. The conclusion about improved hatchability of chickens from eggs collected from young parent stocks are not very strong, since the number of eggs was not high enough and they probably came from the same hatcher.
7. The practicality of microbiota difference between groups should be further explain in relation to chicken growth and development. What advantages/disadvantages for chicken growth and health of those microbiota differences?
Therefore, the paper needs major revision before it can be considered for publication
Author Response
The answer can be found in the attached file.

Reviewer 2 Report
Dear authors,
your study is very interesting and is important for the scientific community and for the industry. Please, find attached my comments.
Best regards, and good luck.

Author Response
Please find our answers in the attached file.

Round 2
Reviewer 1 Report
The paper can be accepted in the present format
Author Response
Dear Reviewer!
Thanks for accepting our revised paper in the present form.
Reviewer 2 Report
Dear Authors,
Your manuscript is very well revised, however, there are minor issues with the incubation temperatures, humidity, and duration (which is one of your experimental factors). Please find attached minor comments.
Best regards.

Author Response
See our answers in the attached file.
